# Two Sides of One Coin: A Comparison of Clinical and Neurobiological Characteristics of Convicted and Non-Convicted Pedophilic Child Sexual Offenders

**DOI:** 10.3390/jcm8070947

**Published:** 2019-06-29

**Authors:** Charlotte Gibbels, Christopher Sinke, Jonas Kneer, Till Amelung, Sebastian Mohnke, Klaus Michael Beier, Henrik Walter, Kolja Schiltz, Hannah Gerwinn, Alexander Pohl, Jorge Ponseti, Carina Foedisch, Inka Ristow, Martin Walter, Christian Kaergel, Claudia Massau, Boris Schiffer, Tillmann H.C. Kruger

**Affiliations:** 1Division of Clinical Psychology and Sexual Medicine, Department of Psychiatry, Social Psychiatry and Psychotherapy, Hannover Medical School, 30625 Hanover, Germany; 2Institute of Sexology and Sexual Medicine, Charité—Universitätsmedizin Berlin, Corporate Member of Freie Universität Berlin, Humboldt-Universität zu Berlin, and Berlin Institute of Health, 10117 Berlin, Germany; 3Division of Mind and Brain Research, Department of Psychiatry and Psychotherapy, CCM, Charité—Universitätsmedizin Berlin, Corporate Member of Freie Universität Berlin, Humboldt-Universität zu Berlin, and Berlin Institute of Health, 10117 Berlin, Germany; 4Department of Forensic Psychiatry, Ludwig Maximilians University Munich, 80336 Munich, Germany; 5Kiel University, Medical School, Institute of Sexual Medicine and Forensic Psychiatry and Psychotherapy, 24105 Kiel, Germany; 6Department of Psychiatry, Otto-von Guericke-University Magdeburg, 39106 Magdeburg, Germany; 7Department of Psychiatry, University of Tübingen, 72076 Tübingen, Germany; 8Division of Forensic Psychiatry, Department of Psychiatry, Psychotherapy and Preventive Medicine, Ruhr University Bochum, LWL University Hospital, 44791 Bochum, Germany

**Keywords:** NeMUP, child sexual offending, child sexual abuse, pedophilia, fMRI, SCID

## Abstract

High prevalence of child sexual offending stand in contradiction to low conviction rates (one-tenth at most) of child sexual offenders (CSOs). Little is known about possible differences between convicted and non-convicted pedophilic CSOs and why only some become known to the judicial system. This investigation takes a closer look at the two sides of “child sexual offending” by focusing on clinical and neurobiological characteristics of convicted and non-convicted pedophilic CSOs as presented in the Neural Mechanisms Underlying Pedophilia and sexual offending against children (NeMUP)*-study. Seventy-nine male pedophilic CSOs were examined, 48 of them convicted. All participants received a thorough clinical examination including the structured clinical interview (SCID), intelligence, empathy, impulsivity, and criminal history. Sixty-one participants (38 convicted) underwent an inhibition performance task (Go/No-go paradigm) combined with functional magnetic resonance imaging (fMRI). Convicted and non-convicted pedophilic CSOs revealed similar clinical characteristics, inhibition performances, and neuronal activation. However, convicted subjects’ age preference was lower (i.e., higher interest in prepubescent children) and they had committed a significantly higher number of sexual offenses against children compared to non-convicted subjects. In conclusion, sexual age preference may represent one of the major driving forces for elevated rates of sexual offenses against children in this sample, and careful clinical assessment thereof should be incorporated in every preventive approach.

## 1. Introduction

With a prevalence of 12.7%, child sexual offenses occur across most ethnic, religious and socioeconomic groups all over the world [1]. This is especially devastating since the effects on children’s wellbeing and development are tremendous and might persist until adulthood [2,3,4]. The high prevalence in comparison to the low conviction rate of child sexual offenders (CSOs) [2] suggests that the majority of CSOs may never get caught. The reasons for the low conviction rate remain unclear but could originate from three different circumstances: (1) Characteristics of the victim (e.g., age), (2) characteristics of the offender (e.g., being better at hiding the crime due to intelligence), and (3) circumstances of the crime (e.g., abuse by an intimate person). One of the most frequently described risk-factors concerning child sexual offending is a pedophilic preference which has been described in about 50% of convicted CSOs [5,6]. The International Classification of Diseases, tenth edition (ICD-10) [7] defines diagnostic criteria of pedophilia as a sexual preference for children—boys or girls or both—usually of prepubertal or early pubertal age including the following criteria: “(A) The general criteria for F65 disorders of sexual preference must be met (G1. Recurrent intense sexual urges and fantasies involving unusual objects or activities. G2. Acts on the urges or is markedly distressed by them. G3. The preference has been present for at least six months). (B) A persistent or a predominant preference for sexual activity with a prepubescent child or children. (C) The person is at least 16 years old and at least five years older than the child or children” [7].

Pedophilia has often been linked to higher psychiatric comorbidities [8,9,10] as well as neurobiological and neuropsychological alterations (e.g., [11]). These results correspond to Krueger and Schiffer [12], who showed that convicted pedophiles serving prison sentences in a forensic treatment facility performed lower on all subtests of a shortened IQ test than a control group except for completing images. Multiple studies showed coherence between pedophilic CSOs and impairment in behavioral inhibition [13,14,15]. Nevertheless, Gerwinn et al. [10] were one of the first revealing an effect of offender status but not of pedophilia on intelligence and other measures in their subgroups. Furthermore, it was shown that essential alterations of brain structure and function—which were assumed to be pedophilia-specific—were predominantly associated with child sexual offending rather than with the pedophilic preference [16,17,18,19]. Most importantly, Kärgel et al. [17] even found non-offending pedophiles to perform significantly better in a behavioral inhibition task (go/nogo paradigm). This leads to the hypothesis that behavioral inhibition potentially secures non-offending pedophiles from acting upon their urges in comparison to sexually offending pedophiles.

Alternatively, there may be effects such as the number of the victims and age of the offender that can lead to imprisonment. Neutze et al. [20] found an age difference between detected and undetected pedophilic CSOs with the detected CSOs being older. Hence, it might be possible that being convicted is just a matter of time. The authors [20] conclude that two sets of factors are associated with detection status: (1) “preceding” ones that might influence whether an individual is detected (e.g., age or education) and (2) “resulting” ones which may either be a response to detection (e.g., emotion-oriented coping, childhood sexual victimization) or reflect a bias in self-report measures (e.g., sexual self-regulation problems, paraphilic interests) in order to avoid negative consequences (e.g., severe punishment) [20].

In the last years, research on neurobiological underpinnings of pedophilia has gained some attention. Unfortunately, the majority of the few studies focusing on child sexual offending and pedophilia mainly examined convicted CSOs. Some studies did not even take sexual preference into consideration at all when comparing groups of offenders and non-offenders. For that reason, results cannot be seen as representative for pedophilia in general, and findings are restricted to the particular group of convicted CSOs [15]. Subsequently, it must be considered that there could be an offender effect rather than an effect of pedophilia itself. Hence, when investigating characteristics due to a pedophilic preference, it is crucial to distinguish between convicted pedophiles, non-convicted pedophiles, and convicted CSOs who are not pedophilic.

Throughout the last years, the research consortium NeMUP, which is the acronym for “neural mechanisms underlying pedophilia and sexual offending against children”, discriminated between pedophilia and child sexual offending by using a two by two factorial design (offender status/pedophilia). Results of the NeMUP consortium showed increased levels of psychiatric comorbidities, sexual dysfunctions, and adverse childhood experiences among pedophiles as well as child sexual offenders [10]. Additionally, regression analyses were more powerful in segregating offender status than sexual preference (mean classification accuracy: 76% versus 68%). In summary, results of the NeMUP consortium showed that executive dysfunctions in CSOs are rather small and may be independent of pedophilic preferences. Neurobiological assessment showed changes in brain structure, metabolism [21] and function, which were particularly associated with the offending status but not with pedophilic preference [16,17,18,19].

Regarding the characteristics of victims that might differentiate between convicted and non-convicted CSOs, it is important to acknowledge that the authors have no intention to blame victims of sexual offenses. Nevertheless, some victim related factors might decrease the likelihood of incarceration of the offenders. It has been shown that children who were older, came from incestuous families, felt greater responsibility for the abuse, and feared negative consequences of disclosure took longer to disclose a crime [22]. Additionally, the sex of the victim could possibly discriminate between convicted and non-convicted CSOs. Since it could be shown that reports of female victims are twice as frequent as of male victims [1] it is also possible that males are victimized equally often but report sexual offenses less frequently [23]. This might be caused by more intense feelings of shame and guilt [24]. Finally, another factor that might differentiate between convicted and non-convicted CSOs may be the quality of the relationship between offender and victim. Whereas it is more probable that sexual offenses against children are committed by a relative or a good friend of the family [25,26] this could also lead to a dilemma concerning a complaint to the police.

While the current investigation did not put an emphasis on victims’ characteristics, it was designed to carefully analyze clinical, neuropsychological and neurobiological features of convicted and non-convicted CSOs in order to unravel possible factors that might increase or decrease the probability to be convicted.

## 2. Method

### 2.1. Participants

Participants of the study were recruited by research associates and practicing psychotherapists as part of the multi-site research project “Neural Mechanisms Underlying Pedophilia and Sexual Offending against Children” (NeMUP, www.nemup.de) which includes five collaborative research sites from (forensic) psychiatry or sexual medicine located in Berlin, Bochum/Essen, Hanover, Kiel and Magdeburg. Most sites of the project were involved in the prevention project “Don’t offend” (www.dont-offend.org) for self-identified pedophiles seeking therapy [27]. Additionally, some subjects were recruited in prisons or during fulfilment of a suspended sentence. Moreover, the official “NeMUP”-website (www.nemup.de) as well as various German internet forums were used to inform self-identified pedophilic men and child sex offenders without pedophilia about the study [10]. For a detailed description of recruitment and collection of the whole NeMUP data, the reader is kindly asked to take a look at Gerwin et al. [10]. The present study only included the NeMUP sample of pedophiles who committed child sexual offenses in the analysis. This group (CSO) contains men who fulfill the criteria of being pedophile/hebephile and have committed at least one “hands-on” [10] delict against children under the age of 14. All participants provided written informed consent before participating, and all local ethics committees located in Berlin, Bochum/Essen, Hanover, Kiel, and Magdeburg approved the study (approval code 6048).

While pedophilia is a sexual preference for prepubescent children and teleiophilia is a sexual responsiveness to adults [10], hebephilia is defined as a sexual preference for children in early pubertal stages. The hebephilic group was subsumed under the label of pedophilia according to ICD-10 that defines pedophilia as a “sexual preference for children…of pre-pubertal or early pubertal stage” (ICD 10).

“Hands-on” delicts were defined as sexual acts that involved touching or manipulating the child’s naked body or genitals (manually or orally) by the offender with the intention of sexually stimulating himself, penetrating the child (anally or vaginally) or forcing the child to (anally or vaginally) penetrate or touch/manipulate the offenders’ genitals [10]. Pedophiles who had not committed such a “hands-on” offense and/or who were currently or historically consumers of material that shows the sexual exploitation of children or so-called indicative pictures (e.g., fully clothed children in erotic poses) were not assigned into this group. This procedure does in no way indicate to degrade such behavior as “non-criminal”, it was used to differentiate between pedophiles who were or were not able to refrain from direct sexual behavior towards children.

In the present study, the group of CSOs was divided into two different subgroups. The first subgroup (non-convicted CSOs) contained men who had not been convicted for their sexual offenses against children, hence pedophiles in the so-called “dark field”. The other group (convicted CSOs) included men that had been convicted for their crimes. Men belonging to the non-convicted CSOs were mainly recruited within the “Don’t Offend” project in Germany [27] and various German internet forums. Participants who were categorized into the subgroup of convicted CSOs were recruited in prisons or during fulfillment of suspended sentences. Exclusion criteria were acute psychiatric or neurological disorders others than pedophilia, acute episodes of alcohol or drug abuse as well as past dependencies and current medication related to sexual functioning. Overall, 31 non-convicted CSOs and 48 convicted CSOs were incorporated into data analysis.

### 2.2. Procedure

Data were collected in two different sessions over two days. The first session included the survey of interviews and diagnostic data, whereas during the second session the magnetic resonance imaging (MRI) assessment was performed.

Structured clinical interview for DSM (SCID) [28] enabled to screen for DSM Axis I and Axis II disorders.

Semi-structured interview was performed in order to gain information about the participants’ sexual interaction interest(s), child pornography consumption, history of offenses in general and biographical information (e.g., age of first coitus, psychiatric disorders in family, own children).

Sexual gender and age orientation was measured by a modification of the Kinsey Scale [29] where participants were asked to indicate range and peak for the age and sex of their preferred sexual partner using the Tanner stages I to V [10]. In case of uncertainty concerning the age preference legal information (if available) and individual case conferences were utilized to ensure a valid clinical diagnosis.

Intelligence was estimated by the German version of the Wechsler Adult Intelligence Scale (WAIS-IV) [30]. Global intelligence was assessed by means of four subtests (“Similarities”, “Vocabulary”, “Block Design”, and “Matrix Reasoning”) [30]. All assessments were carried out by experienced research associates, trained to use these instruments.

Childhood trauma questionnaire (CTQ) [31] was given to the participants to estimate the burden of self-experienced abuse and neglect. The questionnaire screens for a history of five different types of maltreatment (sexual abuse, physical abuse, emotional abuse, physical neglect and emotional neglect). Moreover, an assessment of psychiatric and criminal history of both parents was included in the semi-structured interview for assessing other relevant clinical variables. In the current paper, the difference between the five different types of maltreatment for each group was used to describe differences in experienced childhood trauma. Thresholds for severe childhood trauma can be found at Glaesmer et al. [32].

Empathy was measured using the multifaceted empathy test (MET) [33]. The MET consists of 26 photographs, mostly depicting people in emotionally charged situations. The test assesses cognitive empathy by requiring subjects to deduce mental states of the individuals shown in the photographs. In order to address emotional empathy participants also rate their own emotional reaction in response to the pictures. Unfortunately, no cut-off values for the MET can be found in the literature. However, since the paper focuses on the differentiation between two groups the differences between both groups were used to describe differences in empathy.

Barratt impulsiveness scale (BIS-11) [34] was used to measure self-reported impulsiveness. The questionnaire consists of 30 items and is designed to assess three facets of impulsiveness: (1) attentional impulsiveness, defined as the (in-)ability to concentrate or focus attention, (2) motor impulsiveness, the propensity to act without thinking, and (3) non-planning impulsiveness, or the lack of planning the future and forethought. In the current paper, the difference between the total score for each group was used to describe differences in impulsiveness. For an overview of different clinical scores of BIS-11 the reader is kindly asked to look at Stanford et al. [35].

### 2.3. Imaging Parameters and Processing

Since not all participants fulfilled the criteria to be examined with MRI (e.g., because of claustrophobia or having a pacemaker) only subgroups of the sample were included in the measurements. Therefore, the analysis of the Go/No-go paradigm included 23 non-convicted CSOs and 38 CSOs convicted participants instead of the overall 31 non-convicted CSOs and 48 convicted CSOs.

The MRI images were acquired at five separate 3T MRI scanners that were equipped with 32 channel head coils: 2× Siemens Skyra, 2× Siemens Trio and 1× Phillips Achiva. T1 images were created by means of MPRAGE sequence (slices = 192, FoV = 256 mm, voxel size = 1 × 1 × 1 mm, TR = 2.500 ms, TE = 4.37 ms, flip angle = 7, distance factor = 50%). T2 weighted images were gained using an echo planar imaging (EPI) sequence (slices = 38, field of view = 240 mm, voxel size = 2.3 × 2.3 × 3 mm, time of repetition = 2.400 ms, echo time 30 ms, flip angle = 80, distance factor = 10%). MRI phantom stability measures [36] were accompanied to prevent signal fluctuations across all sites. For functional imaging analysis, the Statistical Parametric Mapping Software (SPM 12) [37] was used. The first five images were discarded to account for T1 relaxation effects. Before statistical analysis, functional volumes were (1) slice time corrected using the middle slice as reference, (2) realigned and unwarped, (3) co-registered to the according T1 image, (4) spatially normalized into Montreal Neurological Institute (MNI) space utilizing the individual T1 image and (5) smoothed with an isotropic Gaussian kernel (full width half maximum of 8 mm).

Go/nogo task: As a measure of impulse control and behavioral inhibition participants performed an event-related go/nogo task preceded by an alertness task of go-trials. The alertness task was applied to allow the participant to familiarize with the MRI environment as well as to provide a baseline measure for reaction time to a simple target stimulus. The projection screen was localized either in front (Hanover) or behind the magnet bore (Berlin, Essen, Kiel). At first, an instruction was presented which informed the participants to respond as fast as possible to any presented stimulus indicated by “X”. Afterwards, the stimulus presentation of 50 alertness trials was initiated. Next, a second instruction screen followed introducing the go/nogo condition containing 150 trials. Participants were asked to respond as fast as possible to frequently presented go-trials (80% indicated by “X”) and withhold the response to infrequently presented no-go trials (20% indicated by “+”). Stimuli were presented using Presentation Software package (Neurobehavioral Systems, Berkeley, CA, USA) and were shown in a pseudo-randomized order, precluding the occurrence of two repeatedly presented no-go stimuli. Stimuli were presented 200 ms with an inter-stimulus-interval (ISI) of 1500–2500 ms. Before scanning, participants were instructed to respond with the right forefinger to the response box. The number of commission errors of no-go trials as well as the reaction times of alertness were used for further statistical analysis to provide a behavioral measure of response inhibition. Outliers differing more than threefold from the distance between the 25% and 75% percentile of the distribution were excluded from analysis. Visual impairments were corrected by a goggle system that was compatible with MR.

### 2.4. Statistical Analysis

Behavioral analysis: SPSS version 24 [38] was used for statistical analysis. To distinguish effects between non-convicted CSOs and convicted CSOs with respect to intelligence, clinical characteristics, empathy, impulsiveness, childhood trauma, victim characteristics, and biographical information, *t*-tests were used for interval data and Chi^2^ tests for dichotomous data. Since the study is part of the NeMUP project there was no specialized sample size calculation for this particular paper. However, the sample size for the whole NeMUP research was calculated for each of the four groups that were included in research (pedophiles with child sexual abuse, pedophiles without child sexual abuse. child sexual abusers without pedophilia and controls). For group one which was included in the current paper, there was a sample size calculation of 60 participants to ensure MRI results were measurable. Concerning non-convicted pedophiles, there are only to papers known by the authors that integrated non-convicted pedophiles into their research. Beier et al. [27] included 53 pedophiles and 22 control participants in his design. Engel et al. [28] included 35 treated pedophiles and 51 treatment refusers in his analysis. For that reason, the sample size of the current paper stands in line with other research published on this topic.

In order to correct for multiple testing, Bonferroni–Holm correction was implemented for different subgroups. Subgroups were: Characteristics of the offender, clinical information, experienced childhood trauma, characteristics of the victims, delict specifics, history of delicts (except hands-on delicts) as well as sexual problems and paraphilias. In order to determine if the number of delicts could be associated with other (sexual) characteristics of the offender, correlations between sexual age preference, sexual gender preference, and quantity of sexual child offenses were included in the analysis. In Table 1 subgroups are described in more detail as well as their descriptive statistic parameters.

For behavioral data of go/nogo paradigm, reaction time, error rate and error quote were analyzed using two sample *t*-tests.

fMRI analysis: SPM12 was used to analyze functional volumes. For first-level analysis the following events were included as regressors in the design matrix: (1) alertness, (2) hits to go-trials, (3) false responses to no-go trials, (4) false responses to no-go trials (commission errors), (5) movement as regressors of no interest. Event-related responses were convolved with the canonical hemodynamic response (HRF), and separate linear contrast images (vectors for each condition) were built. For assessing group differences, contrast images were analyzed using two-sided *t*-tests (non-convicted CSOs vs. convicted CSOs) between groups for the different regressors of interest: (1) alertness, (2) go-trials, (3) successfully inhibited no-go trials and (4) false responses to no-go trials (commission errors). Groups neither differed in age nor IQ so these variables were of no interest for further analysis. The threshold for all analyses was set to *p* = 0.05, familywise error (FWE) corrected for multiple comparisons.

## 3. Results

### 3.1. Characteristics of the Offender

As shown in Table 1, no significant differences could be found which distinguished between the two offender groups concerning age or intelligence (neither in relation to general, verbal nor performance intelligence). Moreover, measurements for empathy or impulsiveness did not separate between non-convicted and convicted CSOs. No correlation between sexual age preference, sexual gender preference, and quantity of sexual child offenses was seen.

### 3.2. Sexuality

Non-convicted and convicted CSOs differed in sexual age orientation χ^2^ (1, *n* = *79*) = 14.00, *p* = 0.03. Convicted CSOs were more often pedophilic (*n* = 19 convicted, *n* = 5 non-convicted) or mixed pedophilic/teleiophilic (*n* = 11 convicted, *n* = 5 non-convicted), whereas non-convicted CSOs showed an age preference more often than mixed hebephilic/teleiophilic (*n* = 3 convicted, *n* = 11 non-convicted). There was no difference in sexual gender orientation χ^2^ (1, *n* = *79*) = 4.01, *p* = 0.135).

### 3.3. Experienced Childhood Trauma

There were no significant differences in self-experienced childhood traumatization as assessed by the CTQ questionnaire between non-convicted and convicted CSOs. A more detailed overview of the data can be found in Table 1.

### 3.4. Clinical Information

As can be seen in Table 1, no significant difference was found between the two groups concerning any clinical features.

### 3.5. Delinquency

CSO delicts: Non-convicted CSOs and convicted CSOs differed in the quantity of their sexual child offenses (*t (78)* = −4.23, *p* < 0.001), but not in the number of victims or in the number of victims by gender. A marginal significance for a difference in the number of male victims was evident (*t (78)* = −2.32, *p* = 0.06). There was neither a correlation between sexual age preference and quantity of sexual child offenses nor between sexual gender preference and quantity of sexual child offenses.

Non-CSO delicts: As can be seen in Table 1, the two groups did not differ concerning delicts other than child sexual offending.

### 3.6. Victims’ Characteristics

The groups showed a marginal significance in the difference of age of female victims (*t (78)* = −2.51, *p* = 0.06).

### 3.7. Go/No-Go Task

Non-convicted and convicted CSOs did not significantly differ in this task, neither in post-error slowing nor in reaction times to commission errors. For detailed information see Table 2.

### 3.8. fMRI Analysis

Successful response-inhibition activated clusters in the anterior cingulate cortex, supplemental motor cortex, insula and middle temporal gyrus (FWE corrected on voxel level, see Table 3) for both groups. No differences were found between both groups.

## 4. Discussion

Overall, only subtle differences between non-convicted and convicted CSOs were identified. Thus, most of the initial hypotheses of the authors concerning differences between convicted and non-convicted pedophilic offenders could not be supported. Data shows that non-convicted CSOs were neither more intelligent nor less impulsive nor suffered less from psychiatric disorders than convicted CSOs in this study. Additionally, no differences could be found concerning their age, the number of victims or the level of empathy. Interestingly, convicted CSOs committed more delicts than non-convicted CSOs. Furthermore, non-convicted and convicted CSOs differed in sexual age orientation. Moreover, all hypotheses except age and gender preference concerning characteristics of victims could not be supported in the present study. However, there was a marginal significance concerning the number of male victims and the age of female victims.

Our data shows that the risk of being caught increases with the number of victims. First, it is likely that with an increasing number of delicts it becomes more probable that one victim is able to identify the offender or to confide in another person about the crime. Second, more frequent delicts of child sexual offending may increase hubris of the offender and lead to carelessness concerning the crime scene, which may finally lead to the offender’s incarceration.

Since convicted CSOs were more often exclusively pedophilic or mixed pedophilic/teleiophilic it is possible that harming a child might lead more often to incarceration than harming an adolescent. This might be due to the social perspective that harming adolescents might be less harmful. Additionally, there is an ongoing debate about whether or not hebephilia is relevant for psychopathology [39]. This debate has been consistent with the evolutionary psychology position that emphasizing the adaptive partner-preference is for fecund females (although females are actually subfecund for one to two years after menarche) [39,40]. Hence it is possible, that mixed hebephilic/teleiphilic preference—which has been shown more often in the group of non-convicted CSOs—will be longer “below radar level” than pedophilic preference. However, since the following data is cross-sectional no causal connections are possible.

The present data suggest that incarceration effects are not as prominent as often stated in criticisms about research with convicted subjects. The general criticism that it is difficult or even impossible to interpret data gained by imprisoned groups exclusively can subsequently be put into another perspective. Nonetheless, it is important to keep incarceration effects, environmental effects, and the effect of critical life events such as being convicted in mind.

Relating to the aim of the study to find differences between both groups which might help to decrease the number of CSOs and—more importantly—undetected CSOs, the difference in the number of delicts between non-convicted CSOs and convicted CSOs actually is of high relevance. However, it is important not to interpret the results as a causal relationship between the number of delicts that would lead to incarceration or not.

The presented data is in line with findings from Neutze et al. [20], showing more group similarities than differences between detected and undetected offenders altogether.

Additionally, the present study compared ‘resulting’ factors of convicted CSOs and non-convicted CSOs. “Preceding” factors are defined as the ones that might influence whether an individual is detected (e.g., age or education) whereas (2) “resulting” ones may either be a response to detection (e.g., emotion-oriented coping, childhood sexual victimization) or reflect a bias in self-report measures (e.g., sexual self-regulation problems, paraphilic interests) in order to avoid negative consequences (e.g., severe punishment). Differences which have been described between convicted and non-convicted CSOs concerning age and education in the study of Neutze et al. [20] (non-convicted CSOs were younger) [20], were not replicated in the present study. Furthermore, the report of suffering more from own childhood victimization in the group of convicted CSO [20] was observable as a marginal significance in the present study. Therefore, the two different sets of factors, proposed by Neutze et al. [20], that are associated with detection status were not observed in the present study. Overall, future research should address these factors.

The consequences of child sexual offending are, as already mentioned, fatal and costly for society (the estimated lifetime economic burden of CSO is approximately $9.3 billion in the US [20]) as well as for children’s wellbeing and development [2,3,4]. Additionally, the devastating effect may persist until adulthood. Thus, emphasis should be laid on preventing offenses. Due to a lack of confidentiality regulations in many countries, offering therapy is only possible after incarceration, i.e., after the crime has not to be concealed anymore. Since therapy may decrease the risk of committing another crime by focusing on dynamic risk factors of offending [27], offering a therapeutic concept to non-convicted CSOs may not only decrease the number of delicts, but it may also reduce the socioeconomic burden and increase the well-being of children. Future research should focus on how preventive approaches may help to reduce the number of offenses.

However, to go one step further offering therapy to non-convicted CSOs might raise the concern that treating this group could lead to better “skills” in hiding crimes and criminal behavior. By working on topics such as behavioral control, social skills, and empathy, non-convicted CSOs might not only learn how to hide a crime better, they may also be able to get better in contacting children and—if so—therapy of non-convicted CSOs might theoretically actively exaggerate the risk of (re)offending. Furthermore, there is a wide discussion in society whether or not offender treatment does more harm than good [41,42]. However, since it is shown in the present study that non-convicted CSOs and convicted CSOs seem to suffer from comparable psychiatric burdens and difficulties, it is unlikely that they will learn skills which may increase their risk of offending. Regarding the fact that detected and undetected CSOs do not differ in clinical characteristics, there is no reason for withholding therapy from undetected CSOs. Moreover, Beier et al. [27] showed that therapy in the German “Dunkelfeld” (dark field) project altered dynamic risk factors for child sexual offending and reduced related behaviors. This knowledge is fundamental to ensure the protection and wellbeing of children and to uncover child sexual abuse as early as possible.

## 5. Limitations

First of all, participants were recruited as part of the multi-site research project NeMUP, which includes five collaborative research sites from (forensic) psychiatry or sexual medicine located in Berlin, Bochum/Essen, Hanover, Kiel and Magdeburg and did not only include convicted and non-convicted CSOs but also healthy controls and “hands-off” pedophiles who have not committed a crime yet. There were several exclusion criteria which were important for other investigations in the NeMUP consortium but might limit generalizing the results of this study. Those were an intellectual disability, psychotic disorder, current severe major depressive disorder (score greater than 15 on the Hamilton Depression Scale) or anxiety disorder (score greater than 25 in the Hamilton Anxiety Scale), clinically predominant substance misuse or dependence and any psychotropic medication. Subsequently, it is possible that differences between CSOs and non-convicted CSOs vanished not because differences did not exist but because of the early exclusion of offenders with a high psychiatric burden. Future research should include participants with current mental disorders and a high psychiatric burden to ensure differences are not disappearing due to strict inclusion and exclusion criteria. Additionally, the questions asked on recruitment might lead to possible bias such as not detecting pedophiles who are ashamed of “coming out” with their sexual preference or not integrating participants with a high psychiatric burden due to exclusion criteria. Since a large number of participants has been recruited within the “Don’t offend”-network this might represent another bias because these subjects are all motivated to undergo psychotherapy.

Another difficulty in the study is the missing detailed data about the crime itself and the crime scene. Due to our quantitative approach, we were not able to include this data as well. However, it is indeed possible that circumstances of the crime and the crime scene are a certain factor for leading to conviction or non-conviction. Circumstances such as sadism or threatening the victim were not included in the analysis. These factors may be important not only for the likelihood of reporting the crime but also for the effort of engaged third parties such as social workers and/or the police. The topic of circumstances of the crime should be taken into consideration for future research. The following paper focuses on the characteristics of the offenders and only a few characteristics of the victims. Since each and every crime has its own characteristics and specifics, it is difficult to investigate the circumstances of a crime by using quantitative research. Concerning alterations in the characteristics of convicted in comparison to non-convicted CSOs, the study only included pedophilic CSOs who are either known by the judicial system, and therefore, convicted or are in the so-called “dark field”.

Moreover, the reliance on self-report concerning detection status of non-convicted CSOs might be a problem for analysis. Future research has to find a way of limiting the risk of false-negative responses by the participants due to the possible withholding of information in virtue of shame for being detected. Another limitation is the confinement of the study sample to individuals who applied voluntarily. For that reason, it is possible that differences or similarities between help-seeking offenders and those not seeking help—independent of detection status—may not have been targeted.

Nevertheless, the study design is cross-sectional, and causal connections are therefore prohibited.

Finally, including measurements for socially desirable responding to the procedure of collecting data would be appreciated.

## 6. Concluding Remark

The study shows effects on a marginally significant level for the number of victims, number of male victims, age of female victims, and self-reported sexual abuse. Additionally, a significant effect for the number of sexual offenses was shown. Hence, convicted CSOs may represent a specific subsample with higher risk and a greater burden of adverse life events though differences appear to be smaller than commonly perceived. The main strength of this study is that it is one of the first focusing on differences between detected and undetected CSOs. Until now, there is only one other study [20] known by the authors that tried to understand which factors may play an important role in the incarceration of CSOs. Because of its design, the study was able to examine differences between those groups.

## Figures and Tables

**Table 1 jcm-08-00947-t001:** Characteristics of non-convicted and convicted child sexual offenders (CSOs).

	Non-Convicted CSOs (*n* = 31)				Convicted CSOs (*n* = 43)				T
	M	SD	Minimum	Maximum	M	SD	Minimum	Maximum	
Offender characteristics									
Age	40.4	10.8	20	59	39.8	9.0	24	62	0.27
Intelligence (WAIS)									
Estimated general IQ	100.26	18.51	62	140	98.27	18.95	66	166	0.46
Estimated verbal IQ	95.00	21.52	54	137	92.14	17.45	57	129	0.65
Estimated performance IQ	106.87	19.41	78.00	143.0	103.31	17.29	63.00	149.00	0.85
Empathy (MET)									
Emotional	5.11	1.25	2.65	7.44	5.33	1.74	1.44	8.11	−0.57
Cognitive	6.98	1.02	4.63	9.00	7.10	1.04	3.88	8.50	−0.49
Impulsiveness (BIS-11)	64.46	9.86	47.00	81.00	63.49	8.59	46.00	88.00	0.44
Family background									
Experienced childhood trauma (CTQ)									
Emotional abuse	11.20	4.90	5.00	23.00	10.24	5.41	5.00	25.00	0.79
Sexual abuse	7.00	2.45	5.00	14.00	9.48	6.34	5.00	25.00	−2.40
Physical abuse	7.70	3.37	5.00	20.00	8.43	4.71	5.00	22.00	−0.74
Emotional neglect	13.20	4.91	5.00	25.00	12.37	4.57	5.00	25.00	0.75
Physical neglect	7.60	2.90	5.00	17.00	7.85	3.05	5.00	20.00	−0.35
Information about the victims									
Age (mean)									
Female victim	10.50	3.19	5.0	15.5	8.68	2.31	5.0	13.0	2.15
Male victim	11.28	2.59	6.0	15.0	10.18	2.48	3.0	14.5	−0.25
History of Delicts (except hands-on Delicts)									
Quantity of violent offense	0.13	0.43	0	2	0.66	1.23	0	2	0.125
Quantity of other offenses	0.83	1.23	0	5	0.55	1.23	0	7	−0.28
Delict specifics									
Quantity of child sexual offenses	1.94	1.41	1	7	4.53	3.91	1	17	−4.23 ***
Number of victims	2.35	2.53	1	11	5.08	8.71	1	60	0.39
Number of female victims	1.58	2.59	0	11	1.33	2.13	0	10	0.48
Number of male victims	0.77	1.43	0	7	3.76	8.81	0	60	−2.32
	*n* ^a^				*n* ^a^				χ^2^
Clinical diagnoses									
SCID I disorders									
Affective disorders	13				16				0.40
Anxiety disorders	8				9				0.63
Sexual disorders	0				0				1.60
Obsessive compulsive disorders	1				1				0.11
Substance disorders	6				12				0.29
Eating disorders	0				1				0.64
ADHD	1				0				1.60
SCID II disorders									0.91
Cluster A	2				3				0.00
Cluster B	11				10				2.22
Cluster C	6				12				0.29
Sexual age orientation									14.00 *
Hebephile	1				3				
Pedophile	5				19				
Mixed pedophile and teleiophile	5				11				
Mixed hebephile and teleiophile	11				3				
Mixed pedophile and hebephile	5				5				
Mixed pedophile, hebephile and teleiophile	4				8				
Sexual gender orientation									4.01
Heterosexual	17				17				
Homosexual	8				23				
Bisexual	6				9				
Intrafamilial (incest) delicts	7				6				1.49
Extrafamilial delicts	24				43				1.49

M: Mean, SD: standard deviation, ^a^ number of participants fulfilling the criteria, * *p* ≤ 0.05, *** *p* ≤ 0.001.

**Table 2 jcm-08-00947-t002:** Inhibited response to no-go trials for both groups.

	Non-Convicted CSOs (*n* = 31)	Convicted CSOs (*n* = 49)	Statistics (*t*-Value)
RT Alertness in ms, M (SD)	263.009 (58.269)	273.540 (47.545)	*t* (78) = −0.74; *p* = 0.46
RT Go in ms, M (SD)	388.062 (43.298)	400.053 (54.644)	*t* (78) = −0.85; *p* = 0.40
Global Error Rate, M (SD)	0.0598 (0.043)	0.0532 (0.0394)	*t* (78) = 0.61; *p* = 0.54
Global Error Quote, M (SD)	11.455 (8.830)	10.632 (7.875)	*t* (78) = 0.37; *p* = 0.71

Coordinates are denoted by x, y, z in mm according to the MNI-space (Montreal Neurological Institute).

**Table 3 jcm-08-00947-t003:** Go/No-Go task performance.

Location (AAL)	Hemisphere	x	y	z	Size	*t*-Value
Inhibited response to no-go trials for both groups, FWE peak level						
Insula	L	−22	24	8	18	6.28
ACC	L	−4	34	22	76	6.15
Middle temporal gyrus	R	46	−42	−8	28	6.03
Middle temporal gyrus	L	−54	−22	−14	16	5.99
Supplemental motor area	R	2	24	46	12	5.43

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
