# Peer review of "Two Sides of One Coin: A Comparison of Clinical and Neurobiological Characteristics of Convicted and Non-Convicted Pedophilic Child Sexual Offenders"

_jcm, 2019, doi:10.3390/jcm8070947_

Round 1
Reviewer 1 Report
The submitted manuscript represents a valuable contribution to the understanding of the potential neurobiological and clinical differences and similarities between two categories of child sexual offenders. The study was performed within the NeMUP consortium. The manuscript does provide a coherent view of the work and the interpretations of the results in the context of prevention and management of child sexual offenses.
My question is whether the authors have any information on the status of familiarity of COSs with the victims, i.e. intrafamilial or extrafamilial offenders? If so, might be these data would offer an added-value to the performed investigation.
In terms of form, several minor corrections and/or clarifications are suggested bellow:
Lines 75-76 - The statement "But what if it is not acting upon their urges but being poor in behavioral inhibition that leads to incarceration?" should be reformulated. Right now, it looks that child sexual offense is a matter of two distinctive paths, either one or another, or the literature does indicate that both paths can be activated in the process of rape or other type of sexual offense.
Lines 94-95 - This is the first time in the body of the manuscript when the authors talk about the NeMUP consortium. I suggest to add a reference for NeMUP and provide a short description of the consortium, specifying that a more detailed description is provided in the Methods section.
Lines 105-107 - I suggest that instead of "...that by no means the authors want to blame victims of sexual offenses." it should read "...that the authors have no intention to..."
Lines 107-108 - In the statement "Nevertheless, some factors might decrease the likelihood of incarceration of the offenders" please specify that the factors are victim-related ones.
Line 121 - please replace the word "accused" with "convicted"
Lines 139-142 - please provide a reference to the definition of "hands-on" delicts.
Line 152 - please provide a reference to the "Don't offend" project in Germany.
Lines 186-190 - Please reorganize the paragraph so that is starts with "Self-reported impulsiveness was measured with..."
Please correct the plural form of "hypothesis" to "hypotheses"- e.g. line 296 is should read "hypotheses", line 301 it should read "all hypotheses".
In Table 1 - please perform a spell check throughout the table, e.g. "quantity of sexual child sexual offenses".
Author Response
We thank the reviewer for the fruitful comments. In the following you will find a point-by-point response to your comments. Thank you very much for your time.
The submitted manuscript represents a valuable contribution to the understanding of the potential neurobiological and clinical differences and similarities between two categories of child sexual offenders. The study was performed within the NeMUP consortium. The manuscript does provide a coherent view of the work and the interpretations of the results in the context of prevention and management of child sexual offenses.
We thank the reviewer for the positive impression his time to review the manuscript and of the manuscript, and appreciate that the reviewer found our manuscript interesting and useful. Below, we have addressed the comments and hope that after a careful revison the manuscript is now suitable for publication.
My question is whether the authors have any information on the status of familiarity of COSs with the victims, i.e. interfamilial or extra familial offenders? If so, might be these data would offer an added-value to the performed investigation.
Indeed, we have information about intrafamilial and extrafamilial delicts. We included two table rows that show the number of intrafamilial and extrafamilial delicts per subgroup and Chi². We did not find differences regarding the family status between both offender groups.
In terms of form, several minor corrections and/or clarifications are suggested bellow:
We would like to thank the reviewer for the remarks. We were able to integrate all of the following corrections and clarifications into our paper.
Lines 75-76 - The statement "But what if it is not acting upon their urges but being poor in behavioral inhibition that leads to incarceration?" should be reformulated. Right now, it looks that child sexual offense is a matter of two distinctive paths, either one or another or the literature does indicate that both paths can be activated in the process of rape or other type of sexual offense.
Thanks you for the remark. We deleted the statement as is indeed misleading the reader.
Lines 94-95 - This is the first time in the body of the manuscript when the authors talk about the NeMUP consortium. I suggest to add a reference for NeMUP and provide a short description of the consortium, specifying that a more detailed description is provided in the Methods section.
We do not have a reference for the Consortium; however we added a link to the webpage and a cross reference to the method section.
Lines 105-107 - I suggest that instead of "...that by no means the authors want to blame victims of sexual offenses." it should read "...that the authors have no intention to..."
We have rephrased this sentence according to the suggestion (see lines 112/113)
Lines 107-108 - In the statement "Nevertheless, some factors might decrease the likelihood of incarceration of the offenders" please specify that the factors are victim-related ones.
We have specified this (see lines ***)
Line 121 - please replace the word "accused" with "convicted"
Cleared
Lines 139-142 - please provide a reference to the definition of "hands-on" delicts.
We have added a reference (see line 144/157)
Line 152 - please provide a reference to the "Don't offend" project in Germany.
We changed the passages and added references according to your suggestions
Lines 186-190 - Please reorganize the paragraph so that is starts with "Self-reported impulsiveness was measured with..."
This paragraph has been reorganized for better readability. The start of the paragraph now reads as: “Barratt Impulsiveness Scale (BIS-11)[34] was used to measure self-reported impulsiveness.”
Please correct the plural form of "hypothesis" to "hypotheses"- e.g. line 296 is should read "hypotheses", line 301 it should read "all hypotheses".
Cleared
In Table 1 - please perform a spell check throughout the table, e.g. "quantity of sexual child sexual offenses".
A spell check has been performed for the table and corrections been made.
Reviewer 2 Report
The presented manuscript entitled: "Two sides of one coin: A comparison of clinical and neurobiological characteristics of convicted and non-convicted pedophilic child sexual offenders" provides with interesting insight into what seems to be a seldom researched area. The authors provide with a novel outlook by comparing convicted vs non-convicted child sexual offenders. The study presents an interesting read, that will probably be of interest to those working in the field of child sex offenders therapy and research.
Overall the manuscript is nicely written, structure and flow seem appropriate. There are some concerns with the study population, population size and also some terminology use in text that is not correct. These ought to be improved before the article may continue to production.
Comments:
Abstract: nicely written, easy to follow. However in line 31 you mention SCID (structured clinical interview), however you have not mentioned this before and I would advise to write it out as some authors may not be familiar with the acronym.
Introduction: Structure is sound. Provides with a good overview of literature. However, the very beginning of the introduction you open with problem that many of the CSO may never get caught and connect this with this fact being devastating for the childs well-being. (p2l46-48) I am sure this is devastating if the cycle of abuse continues, however as the sentences now stand they seem completely unrelated by a logical fallacy. I would suggest this be made more clear.
Methods: I do appreciate that the description of the recruitment was done in another publication, but I believe the readers would appreciate at least a sentence or two about the recruitment process. Especially given the sensitive nature of the topic and the potential bias it may have on the recruited.
Procedure: p4l186: the Barrat Impulsiveness Scale should be bolded in line with other questionnaires.
Statistical analysis: inadequate and should be more elaborated on. You mention correlations, which ones and why? How did you arrive to this sample size? This is only an explorative study, I do not think a sample size calculation is necessary, however given the fairly small sample I wonder on the statistical power. Have you arrived to the desired power? How was this analysed? Have you considered a sample size calculation? What are the sample sizes of similar or other research published?
Results: in the tables it is unclear what the scores mean and in the methods this was not explained. Are there cut-offs? What is "good" what is "bad"? Please either put these info as a footnote under the table or describe it in the methods section for each questionnaire.
p7l263 first time mention of "teleiophilic", this should be explained for those who are not aware of the terminology. Like you did for other terms used in the Procedure section.
Discussion: could benefit for more referencing. Several times in the discussion you mention "trends" that you observed. There are no trends in a cross-sectional analysis as you have no time factor, what you observe are associations. Please use the appropriate terminology in the text. (l303, 338...). Paragraph 305-309 seems to be a strong statement, but you provide no reference. L312-313 I would not explain possible mechanisms by use of mid-20th century novels as there are far more appropriate cultural references, when you have a desire to put one. Furthermore the eroticism of Nabokovs novel is highly discussed and I would prefer this be left to literature scholars. l324-328: this should be clear as this is a cross-sectional study, so no causal connections are possible.
Limitations: study design prohibits causal connections, questions on recruitment may lead to possible bias. Please elaborate.
Otherwise I have no other comments and I hope the authors make the needed corrections in order to make an even better article.
Author Response
We thank the reviewer for his time and the comments that improved our paper. Please find our point-by-point response below:
The presented manuscript entitled: "Two sides of one coin: A comparison of clinical and neurobiological characteristics of convicted and non-convicted pedophilic child sexual offenders" provides with interesting insight into what seems to be a seldom researched area. The authors provide with a novel outlook by comparing convicted vs non-convicted child sexual offenders. The study presents an interesting read, that will probably be of interest to those working in the field of child sex offenders therapy and research.
Overall the manuscript is nicely written, structure and flow seem appropriate. There are some concerns with the study population, population size and also some terminology use in text that is not correct. These ought to be improved before the article may continue to production.
We thank the reviewer for the helpful comments and his time to review the manuscript. Below, we have carefully addressed all issues raised by the reviewer.
Comments:
Abstract: nicely written, easy to follow. However in line 31 you mention SCID (structured clinical interview), however you have not mentioned this before and I would advise to write it out as some authors may not be familiar with the acronym.
We addressed that comment directly and changed our wording in line 31 into “Structured clinical interview (SCID)”
Introduction: Structure is sound. Provides with a good overview of literature. However, the very beginning of the introduction you open with problem that many of the CSO may never get caught and connect this with this fact being devastating for the childs well-being. (p2l46-48) I am sure this is devastating if the cycle of abuse continues, however as the sentences now stand they seem completely unrelated by a logical fallacy. I would suggest this be made clearer.
Thank you for the hint. We restructured the sentences in the following way:
“With a prevalence of 12.7% child sexual offenses occur across most ethnic, religious and socioeconomic groups all over the world [1]. This is especially devastating since effects on children’s wellbeing and development are tremendous and might persist until adulthood [2]–[4]. The high prevalence in comparison with the low conviction rate of child sexual offenders (CSOs) [2] suggests that the majority of CSOs never gets caught.”
Methods: I do appreciate that the description of the recruitment was done in another publication, but I believe the readers would appreciate at least a sentence or two about the recruitment process. Especially given the sensitive nature of the topic and the potential bias it may have on the recruited.
We integrated a more elaborated description of the recruitment process on page 3, lines135-140. “Most sites of the project were involved in the CSO90 prevention project “Don't offend” (www.dont-offend.org) for self-identified pedophiles seeking therapy [27] (Beier et al., 2015) which was used for recruitment. Additionally, some subjects were recruited in prisons or during fulfilment of a suspended sentence. Moreover, the official “NeMUP”-website (www.nemup.de) as well as various German internet forums were used to inform self-identified pedophilic men and child sex offenders without pedophilia about the study[10].”
Procedure: p4l186: the Barrat Impulsiveness Scale should be bolded in line with other questionnaires.
We corrected this.
Statistical analysis: inadequate and should be more elaborated on. You mention correlations, which ones and why? How did you arrive to this sample size? This is only an explorative study, I do not think a sample size calculation is necessary, however given the fairly small sample I wonder on the statistical power. Have you arrived to the desired power? How was this analysed? Have you considered a sample size calculation? What are the sample sizes of similar or other research published?
We appreciate the remarks of the reviewer and have integrated a more appropriate description of our correlations (see page 6, lines 266-267) and sample size calculation (see page3, line 147 to page 4 lines 157). Regarding the correlations the manuscript now reads as: “In order to determine if the number of delicts could be associated with other (sexual) characteristics of the offender, correlations between sexual age preference, sexual gender preference and quantity of sexual child offenses were included into the analysis.”
Regarding the sample size we added the following paragraph: “Since the study is part of the NeMUP project there was no specialized sample size calculation for this particular paper. However, sample size for the whole NeMUP research was calculated for each of the four groups that were included into research (pedophiles with child sexual abuse, pedophiles without child sexual abuse. child sexual abusers without pedophilia and controls). For group one which was included into the current paper there was a sample size calculation of 60 participants to ensure MRI results was measurable. Concerning non-convicted pedophiles there are only to papers known by the authors that integrated non-convicted pedophiles into their research. Beier et al, 2014 [27] included 53 pedophiles and 22 control participants in his design. Engel et al (2018) [28] included 35 treated pedophiles and 51 treatment refusers into his analysis. For that reason the sample size of the current paper stands in line with other research published on this topic.
Results: in the tables it is unclear what the scores mean and in the methods this was not explained. Are there cut-offs? What is "good" what is "bad"? Please either put these info as a footnote under the table or describe it in the methods section for each questionnaire.
We added cut-offs wherever possible into the methods section of our paper.
For CTQ we included: In the current paper the difference between the five different types of maltreatment for each group was used to describe differences in experienced childhood trauma. Thresholds for severe childhood trauma can be found at Glaesmer et al. (2013) [32]. (line 242)
Fr MET we included: “Unfortunately, no cut-off values for the MET can be found in literature. However, since the paper focuses on the differentiation between two groups the differences between both groups were used to describe differences in empathy” (line 250)
For BIS-11 we included the following explanation:
“In the current paper the difference between total score for each group was used to describe differences in impulsiveness. For an overview of different clinical scores in different clinical subgroup (e.g. alcohol abuse) of BIS-11 the reader is kindly asked to look at Stanford et al. (2009) [34].” (line 258)
We have restructured our table and added a footnote to it. Hopefully our results are now more comprehensible.
p7l263 first time mention of "teleiophilic", this should be explained for those who are not aware of the terminology. Like you did for other terms used in the Procedure section.
Thank you for this remark. On page 4 line 158 we explained “teleiophilia” for those who are not aware of the terminology. The sentence introducing sexual age orientation now reads:” While pedophilia is a sexual preference for prepubescent children and teleiophilia is a sexual responsiveness to adults [10], hebephilia is defined as a sexual preference for children in early pubertal stages.”
Discussion: could benefit for more referencing.
Research on this topic is very limited. However, we now tried to include and discuss all relevant literature that is linked to our results.
Several times in the discussion you mention "trends" that you observed. There are no trends in a cross-sectional analysis as you have no time factor, what you observe are associations. Please use the appropriate terminology in the text. (l303, 338...).
Thank you very much for the remark. Instead of the term “trend” we now integrated the term “marginal significance” for a p value of 0.06.
Paragraph 305-309 seems to be a strong statement, but you provide no reference.
We corrected the statement in the following way “Our data shows that the risk of being caught increases with the number of victims” (see line 346/347)
L312-313 I would not explain possible mechanisms by use of mid-20th century novels as there are far more appropriate cultural references, when you have a desire to put one. Furthermore the eroticism of Nabokovs novel is highly discussed and I would prefer this be left to literature scholars.
Thank you for the helpful comment. We deleted the reference to Lolita. In line 357 we pointed out that there is no causal connection possible.
l324-328: this should be clear as this is a cross-sectional study, so no causal connections are possible.
In line 360 we pointed out that there is no casual connection possible.
Limitations: study design prohibits causal connections, questions on recruitment may lead to possible bias. Please elaborate.
We now discussed this point in a more elaborate way (see page 11, lines 427 and page 12, lines 445-446. Specifically we wrote:”Additionally, the questions asked on recruitment might lead to possible bias such as not detecting pedophiles who are ashamed of “coming out” with their sexual preference or not integrating participants with a high psychiatric burden due to exclusion criteria. Since a large number of participants has been recruited within the “Don’t offend”-network this might represent another bias because these subjects are all motivated to undergo psychotherapy.
And: “Nevertheless, the study design is cross-sectional and therefore causal connections are therefore prohibited.” (line 450)
Otherwise I have no other comments and I hope the authors make the needed corrections in order to make an even better article.
Thank you for your and fruitful comments which certainly increase the quality of the paper!
Round 2
Reviewer 2 Report
I would like to thank the authors on their revisions. I would like to congratulate the authors on their work and have no further comments.